# Does *Schistosoma Mansoni* Facilitate Carcinogenesis?

**DOI:** 10.3390/cells10081982

**Published:** 2021-08-04

**Authors:** Verena von Bülow, Jakob Lichtenberger, Christoph G. Grevelding, Franco H. Falcone, Elke Roeb, Martin Roderfeld

**Affiliations:** 1Department of Gastroenterology, Justus Liebig University, 35392 Giessen, Germany; verena.von-buelow@innere.med.uni-giessen.de (V.v.B.); Jakob.J.Lichtenberger@med.uni-giessen.de (J.L.); Elke.Roeb@innere.med.uni-giessen.de (E.R.); 2Institute of Parasitology, BFS, Justus Liebig University, 35392 Giessen, Germany; Christoph.Grevelding@vetmed.uni-giessen.de (C.G.G.); Franco.Falcone@vetmed.uni-giessen.de (F.H.F.)

**Keywords:** *S. mansoni*, schistosomiasis, cancer, carcinoma, HCC

## Abstract

Schistosomiasis is one of the most prominent parasite-induced infectious diseases, causing tremendous medical and socioeconomic problems. Current studies have reported on the spread of endemic regions and the fear of development of resistance against praziquantel, the only effective drug available. Among the *Schistosoma* species, only *S. haematobium* is classified as a Group 1 carcinogen (definitely cancerogenic to humans), causing squamous cell carcinoma of the bladder, whereas infection with *S. mansoni* is included in Group 3 of carcinogenic hazards to humans by the International Agency for Research on Cancer (IARC), indicating insufficient evidence to determine its carcinogenicity. Nevertheless, although *S. mansoni* has not been discussed as an organic carcinogen, the multiplicity of case reports, together with recent data from animal models and cell culture experiments, suggests that this parasite can predispose patients to or promote hepatic and colorectal cancer. In this review, we discuss the current data, with a focus on new developments regarding the association of *S. mansoni* infection with human cancer and the recently discovered biomolecular mechanisms by which *S. mansoni* may predispose patients to cancer development and carcinogenesis.

## 1. Introduction

Schistosomiasis is one of the most common parasitic infectious diseases worldwide, with at least 236 million people requiring preventive treatment in 2019 (WHO 2021) [1]. While the majority of people at risk live in the endemic regions of Africa, *Schistosoma* species are also prevalent in the Middle East, the Caribbean, South America, and Southeast Asia [2]. Schistosomiasis is increasingly being imported into regions with temperate climates by immigrants and travelers from endemic areas [3,4]. Epidemiological case studies of an outbreak of urogenital schistosomiasis in Corsica, France, and the transmission of African schistosomiasis in China underline the potential risk of schistosomiasis spreading into novel areas [5,6]. Two recently published studies analyzed the spread of the intermediate host, a freshwater snail, and zoonotic implications, which might be causative for the current expansion of schistosomiasis [7,8]. Moreover, the development of new hybrid species, which originated from humans via zoonotic spillover from livestock populations, has recently been described in areas where *S. haematobium* is co-endemic with *S. bovis* [9].

After contact with fresh water, these parasitic blood flukes infect their host by penetrating the skin as cercariae, the free-swimming infectious stage of schistosomes [10]. Adult male and female worms live within the venules of their human host, where they mate. Depending on the species, paired schistosomes can produce approximately 300–2000 eggs daily, which are deposited into the stool or urine to reach the environment for continuing their lifecycle [11].

There are three major species infecting humans: *Schistosoma mansoni*, *Schistosoma haematobium*, and *Schistosoma japonicum* [12]. Both *S. mansoni* and *S. haematobium* are present in Africa and the Middle East, while *S. mansoni* also occurs in South America. *S. japonicum* is confined to Asia, mainly China and the Philippines. Schistosomes live an average of 3–10 years but, in some cases, lifespans of nearly 40 years have been reported [13].

Following infection of the final host, schistosome cercariae develop to schistosomulae, the juvenile form of this blood fluke. Schistosomula migrate via the bloodstream to the liver, where they reach the adult stage. Male and female schistosomes pair, and, as couples, they migrate to the urogenital veins (*S. haematobium*) or mesenteric veins of the gut (*S. mansoni*). Following pairing, the male-dependent sexual maturation of the female is completed [14] and egg production starts. *S. mansoni* couples live within the mesenteric veins, where they produce eggs [2]. Chronification of schistosomiasis begins several weeks to months after the cercariae enter their definite host. The symptoms depend on the degree of worm infestation, the rate of oviposition, and the organ site of granulomatous entrapment of the eggs. During chronic stages of infection, half to two-thirds of the eggs are swept away in the circulation to multiple organs instead of being excreted via the stool [2]. The majority of those eggs end up in the liver, causing symptoms of hepatic schistosomiasis [2,11].

Simultaneously to the hepatic manifestation, intestinal schistosomiasis develops when eggs pass through or become trapped in the intestinal tissues [15]. Severe chronic infections with *S. mansoni* are mostly associated with hepatic and intestinal pathogenesis, while urogenital schistosomiasis is mainly caused by *S. haematobium* [10]. The eggs provoke a granulomatous host immune response, which induces chronic inflammation that leads to the pathologic manifestations of schistosomiasis, i.e., portal and pulmonary hypertension, bloody diarrhea, vaginal discomfort, hemospermia, nephropathy, and other organ-specific manifestations [15]. The granulomatous inflammation facilitates the translocation process of the eggs into the gastrointestinal lumen [15]. However, the egg granuloma also protects the host from an exaggerated immune response against the antigenic eggs [2].

In general, infections might initiate or promote carcinogenesis through chronic inflammation due to prolonged persistence of the inducing agent in the host [16]. Amongst other, infections can promote damage of the DNA, proteins, and cell membranes, as well as the modulation of enzyme activities and gene expression [17].

The global burden of cancer was estimated to be 19.2 million new cases and 10 million cancer-related deaths in 2020 [18]. Approximately 20% of human cancers are caused by infectious diseases [16,19]. It was estimated that 0.4% of the new cancers attributable to infections were caused by the trematodes *S. haematobium* (0.3%), *Opisthorchis viverrini*, and *Clonorchis sinensis* (both liver flukes together: 0.1%), which are considered as Group 1 carcinogens by the International Agency for Research on Cancer (IARC) [18]. Infection with the liver flukes *O. viverrini* and *C. sinensis* increases the risk of developing cholangiocarcinoma, while in endemic areas, 46–75% of all bladder cancers can be attributed to *S. haematobium* [20]. Important references to schistosomiasis and cancer date back to the 1940s [21,22,23]. Among the *Schistosoma* species, only *S. haematobium* is classified as a Group 1 carcinogen (definitely cancerogenic to humans), causing squamous cell carcinoma of the bladder, whereas infection with *S. mansoni* is included in Group 3, indicating insufficient evidence to determine its carcinogenicity [24]. In addition, a related liver fluke, *Opisthorchis felineus*, is also classified as a Group 3 agent in a similar manner to *S. mansoni*. Several reports have linked infection with *Opisthorchis felineus* to cancer [25,26]. Interestingly, vaccination against *S. haematobium* has been discussed as an important major public health achievement in preventing cancer, similar to hepatitis B vaccines and HPV vaccines [27]. The question ‘Why does infection with some helminths cause cancer?’ has been intensively discussed and reviewed in the context of *S. haematobium*, *O. viverrini*, and *C. sinensis* [28]. For these three species, even concrete concepts of chemical carcinogenesis, including malignant molecular mediators as well as mechanisms such as the formation of DNA adducts of parasite-released oxysterols and estrogen-like metabolites, have been suggested [29,30,31]. A future-oriented concept might be to study the known carcinogenic mechanisms induced by *S. haematobium*, *O. viverrini*, and *C. sinensis* in models of *S. mansoni* infection.

The data linking *S. mansoni* to cancer are insufficient and, in part, conflicting [32,33]. Nonetheless, case reports and descriptive studies from endemic regions have discussed the possibility of an association between *S. mansoni* infestation and cancer, including hepatocellular carcinoma (HCC) [33,34,35,36,37,38,39,40,41,42], colorectal cancer (CRC) [43,44,45,46,47,48], bladder carcinoma [49], prostate cancer [50], and follicular lymphomas [51,52]. Although *S. mansoni* is classified as a Group 3 carcinogen, the multiplicity of case reports, together with data from animal models and cell cultures, suggests that *S. mansoni* can at least predispose a patient to- or promote cancer. Overall, the epidemiological evidence associating *S. mansoni* infection with cancer is lacking, and studies are often of poor quality or conflicting. Therefore, well-planned, rigorous epidemiological, experimental, and clinical studies are urgently needed to determine the cause–effect relationship between *S. mansoni* and malignancy, and to define the molecular mechanisms involved. In this review, we summarized current data on the association of *S. mansoni* infection with human cancer and reviewed the biomolecular mechanisms by which *S. mansoni* may predispose a patient to cancer development and promote carcinogenesis, as schematically merged in Figure 1.

## 2. Search Strategy and Selection Criteria

For the literature retrieval and study selection process, Medline was searched via the internet using the search engine PubMed (http://www.ncbi.nih.gov/entrez/query.fcgi, accessed on 23 March 2021). The aim of the literature search strategy was to identify case reports and scientific studies of *S. mansoni* infection associated with cancer. The primary search retrieved all studies published between 1980 and 2020 using the following search terms: “*Schistosoma*” OR “*Schistosoma mansoni*” OR “Schistosomiasis” (title and abstract) AND/OR “HCC OR hepatocellular cancer” OR “CRC OR colorectal cancer” (title and abstract) AND “HBV” OR “HCV” OR “IPSE” (title and abstract) AND “1980–2021” (publication date) AND “journal article” (publication type) AND “English OR German” (language). A secondary search was conducted to locate reviews and editorials. The reference lists of all retrieved papers were searched manually to detect additional hits not found by the primary Medline search. All the selected studies had been published as full papers. For the current review, the choice of studies was focused on papers targeting malignancy in the context of *S. mansoni* infection.

## 3. Hepatic Schistosomiasis

Hepatic schistosomiasis results from the host’s granulomatous cell-mediated immune response and the metabolically active and highly antigenic ova of *S. mansoni*. The eggs are swept from the mesenteric veins into the small portal branches of the liver via the portal vein, where they are trapped in the pre-sinusoidal periportal tissues. At the site of egg deposition, an excessive granulomatous response develops, which is harmful to the liver by provoking progressive fibrosis. This can impair the blood flow and thereby induce portal hypertension [12,53]. Progressive fibrosis in the portal tract often leads to obstructive portal lesions and portal hypertension, and might result in hepatomegaly, often in combination with splenomegaly, gastrointestinal bleeding, ascites, hepatic encephalopathy, and liver failure [2]. This severe form of schistosomiasis might be fatal. Hepatic fibrosis displays a wound-healing process, with progressive replacement of functional parenchyma by the extracellular matrix [54,55]. Hepatic stellate cell (HSC) activation and their trans-differentiation into myofibroblasts causes an overproduction of the extracellular matrix (ECM), thus leading to increased vascular resistance, dysregulation of inflammatory responses, and cirrhosis [56,57]. Regarding *S. mansoni* infection, the time from initial infection to advanced fibrosis is usually 5–15 years [12].

HCC represents a fatal outcome of chronic liver disease of different etiologies, which is the fourth-leading cause of cancer-related deaths globally [58]. The risk of HCC depends on hepatic background factors, of which chronic inflammation and fibrosis are major determinants [59]. Patients with liver fibrosis of any etiology belong to the high-risk group for the development of HCC [60,61]. The increase in the incidence of HCC in recent years is partly attributable to the increase in Type 2 diabetes and metabolic syndrome, which can result in non-alcoholic fatty liver disease (NAFLD) and especially in non-alcoholic steatohepatitis (NASH), with or without fibrosis [62,63,64].

The mechanisms involved in oncogene activation, tumor suppressor gene inactivation, chromosomal rearrangement in combination with immune- and inflammatory responses such as the induction of auto-aggressive CXCR6^+^ CD8^+^ T-cells and PD1^+^ CD8^+^ T-cells, telomere shortening, DNA damage, oxidative stress, and autophagy are crucial for HCC development [65,66,67,68,69]. The relationship between HCC and *S. mansoni* has been debated in human cases [32,33]. Most *Schistosoma*-associated HCC cases develop in the presence of advanced chronic liver disease related to chronic hepatitis C virus (HCV) infection, chronic HBV infection, and alcohol abuse [34]. Both HBV and HCV are considered to be major causes of the progression to liver cirrhosis and HCC [70]. Co-infection with *S. mansoni* and HCV or HBV is common in regions where schistosomiasis occurs endemically [71]. The available literature indicates that the co-infection with *Schistosoma* and HBV or HCV likely acts as a cofactor by prolonging the carriage state and resulting more often in chronic hepatitis with fibrosis, as well as higher mortality [34]. A study from Egypt reported on a higher HCC occurrence in patients with co-infection with HCV and *S. mansoni*, suggesting an increased incidence with co-infection [32]. In patients with concomitant HCV and schistosomiasis, HCC was more commonly multifocal and advanced.

Several studies have shown that distinct T cell subpopulations have specific roles not only in immune defense but also in maintaining immune homeostasis. Disturbance of this balance can lead to undesirable side effects such as inflammation, susceptibility to infection, or even autoimmune phenomena, such as the induction of auto-aggressive CXCR6^+^ CD8^+^ T cells and PD1^+^ CD8^+^ T cells in the context of NASH [66,69]. Patients co-infected with *S. mansoni* and HCV exhibited a dominant Th2 response, while the HCV-induced Th1 response was downregulated [72]. Programmed Cell Death Protein 1 (PD-1) plays a vital role in inhibiting immune responses and promoting self-tolerance [73]. As PD-L1 plays an important role in various malignancies, therapeutic modulation of PD-1/PD-L1 signaling is currently being investigated in order to generate novel therapeutic anticancer strategies [73]. Most intriguing, PD-1 signaling is disturbed by *S. mansoni* [74], which may indicate a possible link between *S. mansoni*-promoted carcinogenesis and PD-L1. Moreover, it has been shown that the blockade of PD-1 signaling enhanced the Th2 cell responses and aggravated liver immunopathology in mice infected with *S. japonicum* [75].

There is evidence that the cancer environment generated by the host’s inflammatory cells is a crucial element in the neoplastic process [76]. Infection with *S. mansoni* induces a predictable immunological reaction in the host, a Type 1 immune-dominated response in the acute phase reaction, mainly targeted at worm antigens and characterized by interleukin (IL)-12 and interferon (IFN)-γ [2]. The early Th1 response switches to a Th2-dominated response after the onset of parasite egg production. The release of cytokines such as IL-4, IL-5, IL-10, and IL-13 is a hallmark of the egg-induced granulomatous host immune response driven by Type 2 lymphocytes [77]. *S. mansoni* antigen-induced cytokine production [78], T-cell proliferation, and in vitro granuloma formation involve the activation of protein tyrosine kinases (PTKs) and protein kinase C (PKC) [79]. It has been speculated that immunomodulation might either influence PTKs activity or be the result of altered PTK regulation [80]. Interestingly, SEA-stimulated CD4^+^ T cells from *S. mansoni*-infected patients had a lower proliferation rate than the same cells from the non-infected group [81]. Most importantly, *S. mansoni* treatment reduces HIV entry into cervical CD4^+^ T cells and induces IFN-I pathways [82]. This observation may open a new venue for HIV therapy, as *S. mansoni* infection has been linked with an increased risk of HIV acquisition in women [82]. The authors concluded that the identification of the signaling pathways and mechanisms by which treatment of *S. mansoni* infection could reduce female HIV acquisition is an important step toward designing effective HIV prevention programs [82]. Nevertheless, it is still rather uncertain whether these signal transduction pathways may be involved in *S. mansoni*-associated carcinogenesis. Moreover, recent studies have shown that Th9, Th17, and T follicular helper cells (Tfh) cells might also promote hepatic granulomas and fibrogenesis in schistosomiasis [77]. Additionally, hepatic stellate cells, alternatively activated macrophages, eosinophils, and regulatory T cells have been implicated in the fibrogranulomatous reaction [12]. Regulatory T cells and alternatively activated macrophages in carcinogenesis suppress anti-tumor immune responses and contribute to the development of an immunosuppressive tumor microenvironment, thus promoting immune evasion and cancer progression [83,84].

It is important to emphasize that the schistosome eggs are bioactive entities that proactively interact with the host to achieve their excretion. To this end, the eggs actively manipulate the host’s immune system [2]. The effect of schistosome eggs on host cells can be studied by co-culturing but also by stimulating host cells with egg-conditioned media, soluble egg antigens (SEA), purified biomolecules from the eggs, and recombinantly produced egg proteins. SEA of *S. mansoni* bear hundreds of glycosylated proteins with immunomodulatory potential [85]. Among these, the glycoproteins alpha-1/IL4-inducing principle from *S. mansoni* eggs (IPSE/α1) and omega-1 (ω-1) are the most abundant and require interactions with selective C-type lectins on immune cells [86,87,88]. IPSE/α1 has a C-terminal nuclear localization signal (NLS) that conveys “infiltrin” activity, the ability to infiltrate the nucleus by crossing the cell and nuclear membranes [89,90]. Importantly, IPSE/α1 further contributes to enlargement of hepatic granulomas [91]. IPSE [92] binds to immunoglobulins, with a high affinity for IgE [88]. Once IPSE binds to IgE-bound FceRI receptors on the surface of basophils, it triggers the release of IL-4 and IL-13 [89,93], which directly induces the differentiation of monocytes into the alternatively activated macrophage-like M2 phenotype.

ω-1 is a member of the T2 RNase family, which enters the cell by binding to the mannose receptor on dendritic cells [2], subsequently being internalized into the cell and degrading cellular mRNA and rRNA products. ω-1 has been identified as a powerful Th2-inducing factor. Both the RNase activity and the glycan group are essential for Th2 skewing [87,94].

The identification and characterization of trematode antigens is a relatively young discipline and few data about carcinogenic effects of these antigens have been published yet. Table 1 summarizes the carcinogenesis-associated mechanisms that can be induced by IPSE, a major component secreted from schistosomal eggs, along with a list of the analogous antigens of known Group I carcinogens:

Nevertheless, also other biologically active substances released by *S. mansoni* eggs might influence their carcinogenic potential. It has been shown that *S. mansoni* alters the levels of steroid hormones, which may change the status of the cancer environment by affecting the endocrine system [95].

The firmly established Th2 milieu in chronic schistosomiasis is critical for the subsequent reduction of the T-cell response and immunopathology, and also for the development of fibrosis [96]. Thus, this Th2 milieu might be involved in the progression of malignancy, as Th2 cytokines are related to cancer growth or metastasis [97,98]. Regulatory B-cells (Bregs) accelerate HCC formation by induction of the growth and migratory potential of cancer cells [99]. Schistosome egg antigens, including the glycoprotein IPSE/α1, trigger the development of Bregs [100]. Additionally, Bregs are capable of producing IL-10, which induces Treg differentiation, thereby supporting a tolerogenic microenvironment [100]. While inflammatory injury drives both fibrogenesis and carcinogenesis, the tolerogenic microenvironment of the liver conveys immunosuppressive effects that encourage cancer growth [101]. In particular, the prevalence of Tregs is strongly correlated with HCC progression [102]. In schistosomiasis, regulatory T-cells (Tregs) exert an immunosuppressive role to limit the granulomatous inflammation and fibrosis [77].

The dominant Th2 response of *S. mansoni* infection might have therapeutic activity, e.g., reducing the incidence and development of diabetes in NOD mice [103]. Metabolic changes correlated with the eggs deposited in the liver have been shown in *S. japonicum*-infected mice, indicating the promotion of glycolysis-related genes on the one hand and downregulation of gluconeogenesis-related genes on the other hand [104]. In addition to the metabolic changes induced by the schistosome ova, SEA have the potential to alter cancerogenic signaling at the molecular level in vitro and in vivo [105,106]. In cancer progression, metabolic reprogramming, oxidative stress, and DNA damage display crucial events. Whether *S. mansoni* SEA induce metabolic changes in the liver or even a reprogramming of the hepatic glucose metabolism remains ambiguous. However, it is tempting to speculate that the eggs benefit from the host’s metabolic environment to ensure their own survival, thereby disturbing the host’;s metabolic balance.

Studies on animal models have demonstrated amelioration of arthritis [107,108], colitis [109,110,111], Type 1 diabetes [112], and sepsis [113] with concomitant schistosome infection through induction of Th2-specific and immune modulatory cytokines IL-4, IL-5, IL-13, and IL-10 and suppression of Th1-specific cytokines IL-1β, IL-6, IL12, TNFα, and IFNγ. In this context, it is interesting to note that the protective effect of *S. japonicum* infection in an arthritis model is infection stage-dependent and was attributed to the Th2-dominated response after the onset of parasite egg production, while the early Th1 response in schistosomiasis exacerbated arthritis [108,114]. In addition, *S. mansoni* can induce Foxp3^+^ Tregs, tolerogenic dendritic cells (DCs), and alternatively activated macrophages, which provide protective effects in arthritis [115], colitis [111,116], autoimmune diseases [117], Type 1 [112,118] and Type 2 diabetes [119,120], and sepsis [121].

Although the above mentioned positive immune modulatory effects of schistosomiasis and schistosomal products have been shown for different diseases, the Th2 milieu, Tregs, and tolerogenic DCs are known to suppress tumor-specific immune responses, thus establishing an immunosuppressive tumor microenvironment [83], which may account for the procarcinogenic properties of *S. mansoni* infections.

The role of *S. mansoni* as a risk factor for developing HCC was studied in a mouse model by inducing HCC with diethylnitrosamine (DEN) and concomitant schistosomiasis [122]. The cancerogenic effect of DEN was enhanced by *S. mansoni*. This led to the conclusion that *S. mansoni* is able to accelerate dysplastic changes in the presence of another risk factor, thus promoting cancer development, which appeared to be more aggressive in the presence of *S. mansoni* [122]. However, in this early study, it became clear that data are missing to explain how infection with *S. mansoni* may predispose patients to HCC development and how *S. mansoni* alters cancer growth, angiogenesis, and metastasis. In the liver of DEN-treated mice, infiltrating macrophages may facilitate the initiation of cancer development by the release of TNF-α and IL-6, which are able to activate essential morbidity-associated nuclear factors and signaling pathways such as NF-κB and STAT3 in HCC progenitor cells [114,123]. Aberrant NF-κB and STAT3 signaling is important for cell survival, and it has been involved in the pathogenesis of most human malignancies [124,125]. In this context, it has been demonstrated that *S. mansoni* infection caused mitochondrial damage, resulted in the release of ROS and superoxide, and upregulated NF-κB (p65) expression [126]. Thus, it appears likely that *S. mansoni*-dependent activation of NF-κB signaling in DEN-treated mouse livers may provide cancer cells with a survival advantage by the forced induction of anti-apoptotic genes.

Macrophages and eosinophils generate free radicals and nitrogen species in response to parasites [127]. These products can oxidize and damage DNA, induce DNA mutations, and may lead to genetic instabilities and malignant transformation [116]. Dysregulation of the enzymatic antioxidant system has been observed in a *S. mansoni*-infected mouse model. This was accompanied by the significantly lower activity of superoxide dismutase but increased catalase activity in the liver [128].

An Egyptian study with subjects infected with *S. mansoni*, *S. haematobium*, or both in parallel analyzed the most frequent mutations in human HCC [129]. A mutation in codon 249 of the p53 gene was identified in populations exposed to a high dietary intake of aflatoxin B1(AFB1) [129]. *S. haematobium* is known to cause mutations in the p53 gene. The combination of schistosomiasis and aflatoxin B1 increased the incidence of p53 gene mutations [129]. The mutations in codon 249 of the p53 gene were increased in patients infected with *S. haematobium* compared with those infected with *S. mansoni* or a combination of both species and compared with control subjects [129]. No p53 gene mutation was detected in hepatic DNA from schistosomiasis-free patients [129]. Significant amounts of N7-guanine-AFB1 adducts and novel adenine adducts (*p* < 0.01) were detected in patients with schistosomiasis, mostly in patients infected with *S. haematobium* or a combination of both species, suggesting that schistosomiasis and exposure to aflatoxin B1 act synergistically to increase the incidence of p53 gene mutations [129]. The authors concluded that the increase in p53 mutations might enhance the progression of HCC at an early age in patients with schistosomiasis [129]. Discrete mutation profiles that differ between liver fluke-associated cholangiocarcinoma and non-liver fluke-associated cholangiocarcinoma *have been described* [130]. In this regard, the identification of mutational signatures for HCC with concurrent infection with *S. mansoni* might be a promising approach to gain further insights into the mechanisms that might link *S. mansoni* to HCC.

Data from a hamster model, cell culture studies, and human biopsies provided evidence that antigens released from the *S. mansoni* ova are able to modulate host-specific cancerogenic pathways by activation of c-Jun and STAT3, correlating with cell-cycle activation and DNA double-strand breaks [106]. In vitro experiments with primary hepatocytes and Huh7 revealed that the activation of c-Jun and STAT3 as well as DNA repair were induced by soluble egg antigens (SEA) and egg-conditioned medium, and, in particular, by IPSE/ α1. STAT is constitutively expressed in cancer cells and is involved in cancerogenesis and survival [131]. Furthermore, the constitutive activation of STAT3 inhibited the maturation of dendritic cells through enhanced expression of IL-10 and VEGF. Blocking STAT3 resulted in the activation of both the innate and adaptive anti-tumor immune responses [132]. c-Jun is a major regulator of survival and proliferation during hepatic regeneration [133,134]. In vivo studies have underlined the importance of c-Jun for the induction and survival of liver cancer [135,136]. The HCV core protein potentiated chemically induced HCC through c-Jun and Stat3 activation in transgenic mice, which, in turn, enhanced cell proliferation, suppressed apoptosis, and impaired oxidative DNA damage repair, finally leading to hepatocellular transformation [137]. The permanent activation of hepatocellular carcinoma-associated proto-oncogenes such as c-Jun and the associated transcription factors, including Stat3, by substances released from tissue-trapped schistosome eggs may represent important factors contributing to the development of liver cancer, e.g., in HBV or HCV patients with concomitant *S. mansoni* infection [106].

Schistosome-induced angiogenesis has been described in human studies and experimental models. During murine infection, vascularization was found to be significantly enhanced in regions with a high egg concentration. In addition, the protein level of the pro-angiogenic factor VEGF was significantly higher in human subjects. Inhibition of angiogenesis by endostatin in infected mice reduced hepatic egg deposition, worm load, and mRNA expression of hepatic VEGF [138,139]. Studies identified living eggs and SEA as the inducers of endothelial proliferation [140], which may promote angiogenesis within hepatic granulomas by upregulating endothelial cell VEGF. The conditions created by vessel occlusion, such as hypoxia, acidic aid pH, and low glucose concentrations, may also contribute to the observed neovascularization [141]. Similar to what has been observed in cancer, the growth of new vessels would maintain blood flow in scenarios of vessel occlusion, which would enable the recruitment of leucocytes to developing granulomas and ensure an adequate supply of oxygen and nutrients at these sites [15].

Figure 2 summarizes the known biomolecular factors and pathways induced by *S. mansoni* that alter cellular processes such as proliferation, apoptosis, or DNA damage that are linked to malignant hallmarks such as immune escape, survival, or tumor growth.

## 4. Intestinal Schistosomiasis

Globally, colorectal cancer (CRC) is the third most commonly diagnosed malignancy and the second leading cause of cancer death [142]. Advanced colon polyps are strong risk factors for colorectal cancer [143]. The primary presenting symptoms of intestinal schistosomiasis are usually tenesmus and the rectal passage of blood and mucous and bloody diarrhea [144]. Egg deposition and granuloma formation eventually lead to acute then chronic schistosomal colitis and are commonly associated with polyp formation [145,146,147,148]. The preferential site of schistosomal polyps is the rectum, followed by the sigmoid colon [149]. Histopathological evaluation of the polyps revealed granulomatous inflammation with multiple, mostly calcified *S. mansoni* eggs in the center [145,146,147,150]. Several case reports have described the associations of *S. mansoni* with prostate adenocarcinoma and colorectal cancer [43,45,46,50]. It has been shown that *S. mansoni*-associated colorectal cancer is characterized by a high percentage of synchronous tumors and mucinous adenocarcinomas, and a higher frequency of advanced Stage III and IV tumors [46]. Schistosomal CRC (SCRC) shares unique characteristics independently of the *Schistosoma* species:SCRC occurs at a younger age—6–16 years earlier than ordinary CRC [21,46,151,152,153]—which could be due to early environmental exposure to schistosomal infections in childhood. Preschool-aged children may already have been exposed to this disease [154].SCRC incidence is consistently higher in males [46,151,153]. This male predominance was related to greater employment in agricultural work, and higher rates of contact with water, such as in workers busy with car washing [155].SCRC appears preferentially in the rectum, with a mucinous histology [46,156,157].

Contrarily, one study demonstrated that treatment with *S. mansoni* antigens reduced the number of tumors and the diameter of 1,2-dimethylhydrazine-induced CRC in mice [158]. The authors hypothesized that the protective effects resulted from a *S. mansoni* antigen-induced non-specific immune response or a cross-reactive adaptive immune response induced by *S. mansoni* glycosylated antigens [158].

There are only a few data available about the molecular mechanisms of intestinal schistosomiasis from *S. mansoni* infection analyzing the dysregulation of the malignant colon. A recent study described a DNA repair defect and a RAS mutation in two patients with *S. mansoni*-associated CRC [159]. The authors discussed whether these mutations might have been crucial for carcinogenesis or mere coincidence [159]. Several studies indicated the involvement of p53 [46,48,160]. Madbouly et al. found a significantly higher expression of p53 in SCRC compared with non-*S. mansoni*-infected CRC patients [46]. In contrast, in another study with 75 Egyptian CRC specimens, Zalata et al. found similar expression patterns of p53 and c-Myc in both groups, but significantly more SCRC patients were Bcl-2 positive compared with CRC patients without *S. mansoni* infection [48]. Bcl-2 was positivity correlated with greater apoptotic activity in cancers of the non-infected group. Zalata et al. concluded that the genotoxic agents produced endogenously during *S. mansoni* infection might be involved in the pathogenesis of CRC, with the overexpression of Bcl-2 leading to a reduction in programmed cell death in potential latent cancer foci [48]. In the study of Nacif-Pimenta et al., the regulatory effects of eggs from *S. mansoni* and *S. haematobium* on epithelial cell lines from the urinary and biliary tract were investigated in vitro [160]. It was demonstrated that *S. haematobium* eggs had a higher potential to induce proliferation in epithelial cells [160]. Nevertheless, gene expression analysis of oncogenes in human urothelial cells confronted with both schistosome species revealed that only *S. mansoni* eggs induced a significant dysregulation of the CRC signaling pathways, including the upregulation of TNF, RUNX1, the proto-oncogenes c-Myc and c-Jun, NF-κB1, Scr, and Bcl-2 [160]. In contrast, p53-associated pathways were downregulated by both *S. mansoni* and *S. haematobium* eggs [160]. Previously, we reported the upregulation of the JNK/c-Jun signaling pathway in the liver of *S. mansoni*-infected hamsters, human liver samples, and primary liver cells exposed to *S. mansoni* SEA [106]. Interestingly, a similar activation pattern of the proto-oncogene c-Jun was discovered in epithelial colon cells and colon specimens exposed to *S. mansoni* SEA [105]. Urothelial cells co-cultured with either *S. mansoni* eggs or *S. haematobium* eggs resulted in the upregulation of WNT5a and WNT5b, which have been implicated in CRC via the non-canonical WNT signaling pathway [161]. Activation of the non-canonical WNT signaling pathway was also found in epithelial colon cells exposed to SEA [105]. Hence, it can be speculated that the carcinogenicity of chronic infection with schistosomes may not only depend on the schistosome species but also on the host tissue exposed to SEA.

## 5. Conclusions

Egg-triggered processes such as the activation of protooncogenes, Th2-immune modulation, and the tolerogenic reprogramming of DC and Tregs may be molecular triggers for *S. mansoni*-promoted carcinogenesis. Clinical data including more extensive cohort studies about affected patients may allow better insights into the association of *S. mansoni* with HCC and CRC. Globalization, the expansion of endemic areas, the justified fear of developing resistance to praziquantel, and the association with malignancy may underline the urgency to find new ways to control schistosomiasis.

## Figures and Tables

**Figure 1 cells-10-01982-f001:**
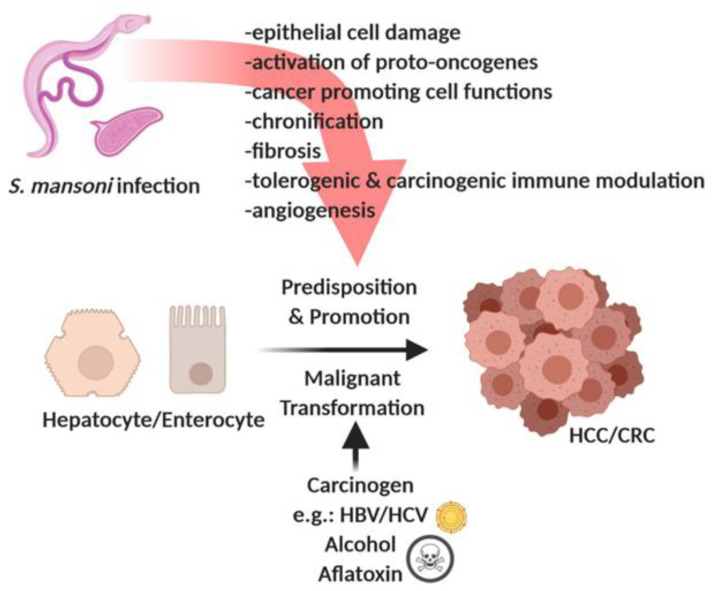
Infection with *S. mansoni* induces processes predisposing a patient to malignant transformation and/or promoting hepatocellular carcinogenesis (HCC) and/or colorectal carcinogenesis (CRC). Created with BioRender.com, online link: https://biorender.com/, (accessed on 25 March 2021).

**Figure 2 cells-10-01982-f002:**
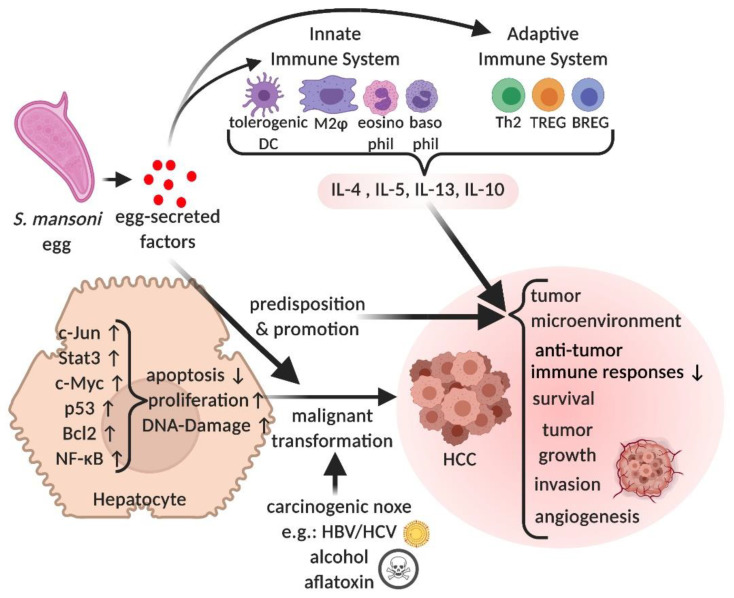
Egg-secreted factors induce tolerogenic and cancer-promoting immune modulations, i.e., the suppression of anti-tumor immune responses, and contribution to the development of an immunosuppressive tumor microenvironment. Egg-secreted factors trigger cellular processes that may predispose hepatocytes to neoplastic transformation or promote malignancy. Created with BioRender.com, online link: https://biorender.com/ (accessed on 25 March 2021).

**Table 1 cells-10-01982-t001:** Table summarizing the identified parasite-derived proteins with suspected pro-carcinogenic activity. BLCA, bladder cancer; CC, cholangiocarcinoma; CRC, colorectal carcinoma; HCC, hepatocellular carcinoma; SCC, squamous cell carcinoma.

Molecule	Effect	Type of Cancer Associated with Infection	Expressing Lifecycle Stage	Reference (PMID)
*Opisthorchis viverrini*				
Granulin (*Ov*-GRN-1)	angiogenesis		eggs, metacercariae, juveniles, adults	25450776
	wound healing	CC		26485648
	proliferation			19816559
*Clonorchis sinensis*				
CsGRN	cell migration and invasion	CC, HCC	adults	28545547
Csseverin	anti-apoptotic		metacercariae and adults	24367717
*Schistosoma haematobium*				
IPSE/α1	proliferation and angiogenesis	BLCA, SCC	eggs only	33101456
*Schistosoma mansoni*				
IPSE/α1	proliferation, c-Jun and STAT3 activation	HCC	eggs only	30053321
		CRC		33361772

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
