# Peer review of "Does Schistosoma Mansoni Facilitate Carcinogenesis?"

_cells, 2021, doi:10.3390/cells10081982_

Round 1
Reviewer 1 Report
The essay and review paper by von Bülow and co-authors addresses the prospect that infection with Schistosoma mansoni might be more seriously considered to be a biological carcinogen, specifically in the context of bowel cancer and/or liver cancer. Three other trematode infections, infection with Schistosoma heamatobium, infection with Clonorchis sinensis, and infection with Opisthorchis viverrini have already been classified as Group I carcinogens by the International Agency for Research on Cancer. In addition, infection with a related liver fluke, Opisthorchis felineus, is a Group 3 agent (e.g., Gouveia et al 2017 Carcinogenesis 38(9):929-937 PMID: 28910999; Pakharukova et al 4open 2019, 2, 23) in a similar fashion to S. mansoni.
The manuscript provides an objective and comprehensive review of the literature on this issue from 1890 to 2020. In addition, the early paragraphs including Figure 1 outline the infection, natural history, and pathogenesis and illustrate some possible routes to malignancy as the result of the schistosome infection including epithelial cell damage from the passage of eggs across the bowel, hepatic fibrosis, angiogenesis in the vicinity of entrapped eggs, and others.
The details discussed relating to the authors’ own research on S. mansoni-infected hamsters, cell cultures, and human biopsies, Roderfeld et al 2020 Hepatology 72, 626-641, provide intriguing links to malignancy including activation of the c-Jun and STAT3 associated pathways as the consequence of antigens released from the egg of S. mansoni. As the authors relate, there are compelling parallels in the findings to established links between the core protein of hepatitis C virus (HCV), a Group 1 biological carcinogen, its effects on c-Jun and STAT3 activation, proliferation, suppressed apoptosis, impaired DNA-adduct mutational repair, and malignant transformation of hepatocytes.
The authors might care to discuss the possibility of mutational signatures. With cholangiocarcinoma caused by infection with fish-borne flukes, particularly O. viverrini, discrete mutation profiles have been described that differ between liver fluke associated CCA and non-liver fluke associated CCA (Jusakul et al 2017 Cancer Discovery 7(10):1116-1135, doi: 10.1158/2159-8290.CD-17-0368. PMID: 28667006). Might such an approach be considered with respect to, for example, HCC where concurrent infection with S. mansoni has taken place?
Minor points
Italicize S. mansoni in several places.
Consider the inclusion of a table of key antigens released by the schistosome egg, and their effects, along with a list of analogous antigens of known Group I carcinogens, including for example granulin of O. viverrini; with key citations.
Reviewer 2 Report
The review article by Verena von Bülow et al. entitled” Does Schistosoma mansoni facilitate carcinogenesis?” evaluated the possible mechanisms of Schistosoma mansoni-induced carcinogenesis. It is a well-written article with proper references cited. If the article can address several more topics that will make this manuscript more valuable.
- The authors addressed possible mechanisms involved in egg-secreted factors-induced a tolerogenic and cancer-promoting immune modulation. Information on the signal transduction pathways lacked. Authors may want to describe signal transduction pathways involved in S mansoni-induced carcinogenesis.
- S. haematobium and S. mansoni are classified as carcinogens in different groups. Is there any way to compare the mechanisms of those two agent-induced carcinogenesis?
- Schistosoma mansoni’s infection causes unbalance of hormones, especially steroid hormone. Those unbalanced hormones may change status of cancer environments. Authors may need to address it.
- PD-L1 has been shown to affect cancer proliferation. Change in immune causes increasing in PD-L1. Schistosoma mansoni’s infection induces immunomodulation. Is it possible linkage between Schistosoma mansoni-induced carcinogenesis and PD-L1?
-
Author Response
The review article by Verena von Bülow et al. entitled” Does Schistosoma mansoni facilitate carcinogenesis?” evaluated the possible mechanisms of Schistosoma mansoni-induced carcinogenesis. It is a well-written article with proper references cited. If the article can address several more topics that will make this manuscript more valuable.
- The authors addressed possible mechanisms involved in egg-secreted factors-induced a tolerogenic and cancer-promoting immune modulation. Information on the signal transduction pathways lacked. Authors may want to describe signal transduction pathways involved in S mansoni-induced carcinogenesis.
Author’s reply: We thank the reviewer for this suggestion and introduced the following passage into the revised manuscript: lines 214-227 ´S. mansoni antigen-induced cytokine production [78], T-cell proliferation and in vitro granuloma formation involve the activation of protein tyrosine kinases (PTKs) and protein kinase C (PKC) [79]. It was speculated that immunomodulation either might influence PTKs activity or might be the result of altered PTKs regulation [80]. Interestingly, SEA-stimulated CD4+ T cells from S. mansoni-infected patients have a lower proliferation rate than the same cells from the non-infected group [81]. Most important, S. mansoni treatment reduces HIV entry into cervical CD4+ T cells and induces IFN-I pathways [82]. This observation may open a new venue for HIV-therapy as S. mansoni infection has been linked with an increased risk of HIV acquisition in women [82]. The authors concluded that the identification of signaling and mechanisms by which treatment of S. mansoni infection could reduce female HIV acquisition is an important step toward designing effec-tive HIV prevention programs [82]. Nevertheless, it is yet rather uncertain whether these signal transduction pathways may be involved in S. mansoni-associated carcinogenesis.´
- S. haematobium and S. mansoni are classified as carcinogens in different groups. Is there any way to compare the mechanisms of those two agent-induced carcinogenesis?
Author’s reply: It is of great importance to examine if known malignant mechanisms of the group I trematodes have an analogue impact in S. mansoni infection. Appropriate experiments with parallel infections or stimulation in the same setting may help to answer this question. Therefore, we included the following passage into the revised version of our manuscript, line 113-115: ´A future-oriented concept might be to study the known carcinogenic mechanisms induced by S. haematobium also in models for S. mansoni infection.´
- Schistosoma mansoni’s infection causes unbalance of hormones, especially steroid hormone. Those unbalanced hormones may change status of cancer environments. Authors may need to address it.
Author’s reply: We included the following sentences to address this point: lines 265-268 ´Nevertheless, also other biologically active substances released by S. mansoni eggs might influence the carcinogenic potential. It has been shown that S. mansoni alters the levels of steroid hormones which may change the status of cancer environment by affecting the endocrine system [95].´
- PD-L1 has been shown to affect cancer proliferation. Change in immune causes increasing in PD-L1. Schistosoma mansoni’s infection induces immunomodulation. Is it possible linkage between Schistosoma mansoni-induced carcinogenesis and PD-L1?
Author’s reply: This is a very important aspect that is addressed in the revised version of our manuscript. We included the following sentences to discuss this point: lines 198-206 ´Programmed Cell Death Protein 1 (PD-1) plays a vital role in inhibiting immune responses and promoting self-tolerance [73]. As PD-L1 plays an important role in various malignancies, therapeutic modulation of PD-1/PD-L1 signaling is currently investigated in order to generate novel anticancer therapeutic strategies [73]. Most intriguing, PD-1 signaling is disturbed by S. mansoni [74] which may indicate a possible link between S. mansoni-promoted carcinogenesis and PD-L1. Moreover, it has been shown that the blockade of PD-1 signaling enhances the Th2 cell responses and aggravates liver immunopathology in mice infected with S. japonicum [75].´
Reviewer 3 Report
Dear Verena von Bülow and colleagues,
I have enjoyed reading your review about the link between Schistosoma infestation and cancer. I have no comments and just some recommendations:
- To create a list of acronyms at the beginning of the review.
- To include other charts or drawings to summarize the outcomes of your literature research breaking plain pages
That is all, congratulations.
Round 2
Reviewer 2 Report
The revised manuscript was well-reorganized. Authors responded to most of comments. It is a novel idea to discuss the biomolecular mechanisms involved in S. mansoni-predisposed cancer development and carcinogenesis. It is recommanded to be accepted as it was.